# Cutting-Edge CAR Engineering: Beyond T Cells

**DOI:** 10.3390/biomedicines10123035

**Published:** 2022-11-24

**Authors:** Luisa Chocarro, Ester Blanco, Leticia Fernández-Rubio, Hugo Arasanz, Ana Bocanegra, Miriam Echaide, Maider Garnica, Pablo Ramos, Sergio Piñeiro-Hermida, Ruth Vera, Grazyna Kochan, David Escors

**Affiliations:** 1Oncoimmunology Unit, Instituto de Investigación Sanitaria de Navarra (IdiSNA), Navarrabiomed-Fundación Miguel Servet, Universidad Pública de Navarra (UPNA), Hospital Universitario de Navarra (HUN), 31008 Pamplona, Spain; 2Division of Gene Therapy and Regulation of Gene Expression, Instituto de Investigación Sanitaria de Navarra (IdISNA), Cima Universidad de Navarra, 31008 Pamplona, Spain; 3Medical Oncology Unit, Instituto de Investigación Sanitaria de Navarra (IdiSNA), Hospital Universitario de Navarra (HUN), 31008 Pamplona, Spain

**Keywords:** CAR-myeloid, CAR-macrophage, CAR-monocyte, CAR-NK, immunotherapy, tumour microenvironment

## Abstract

Chimeric antigen receptor (CAR)-T adoptive cell therapy is one of the most promising advanced therapies for the treatment of cancer, with unprecedented outcomes in haematological malignancies. However, it still lacks efficacy in solid tumours, possibly because engineered T cells become inactive within the immunosuppressive tumour microenvironment (TME). In the TME, cells of the myeloid lineage (M) are among the immunosuppressive cell types with the highest tumour infiltration rate. These cells interact with other immune cells, mediating immunosuppression and promoting angiogenesis. Recently, the development of CAR-M cell therapies has been put forward as a new candidate immunotherapy with good efficacy potential. This alternative CAR strategy may increase the efficacy, survival, persistence, and safety of CAR treatments in solid tumours. This remains a critical frontier in cancer research and opens up a new possibility for next-generation personalised medicine to overcome TME resistance. However, the exact mechanisms of action of CAR-M and their effect on the TME remain poorly understood. Here, we summarise the basic, translational, and clinical results of CAR-innate immune cells and CAR-M cell immunotherapies, from their engineering and mechanistic studies to preclinical and clinical development.

## 1. Introduction: Adoptive T Cell Therapy (ACT)

Immunotherapy has recently emerged as an effective cancer treatment, generating durable and successful responses. Two main types of immunotherapies have emerged: immune checkpoint inhibitors and adoptive cell therapies [1]. Immune checkpoint blockade immunotherapies are the most widespread, generally consisting of the administration of monoclonal antibodies that block co-inhibitory interactions between T cells and cancer cells [1]. This type of immunotherapy is based on the blockade of T cell inhibitory receptors such as programmed death-1 (PD-1) or cytotoxic T lymphocyte antigen 4 (CTLA-4) [2]. The most commonly used antibodies in clinical practice are nivolumab, pembrolizumab and ipilimumab [3,4,5]. Although effective, a large number of patients show primary or acquired resistance to treatment or even develop hyperprogressive disease with fast clinical deterioration [6,7].

In this context, adoptive T cell therapy (APC) has emerged as a promising therapeutic strategy to treat cancer that might overcome the clinical limitations of immune checkpoint blockade immunotherapies [8,9,10]. The most effective APC strategies consist of the genetic modification of autologous or allogeneic T cells to express tumour antigen-specific T cell receptors (TCRs) and chimeric antigen receptors (CARs). Gene modification is usually carried out either with retrovirus or lentivirus vectors. These T cells exert their cytotoxicity upon antigen contact independently of the major histocompatibility complex (MHC) engagement restriction [8,11,12,13,14].

Chimeric antibody T cell receptor (TCR) molecules were first designed and described by Gross, Alaks and Eshhar in 1989 [15]. These molecules target tumour antigens through a single-chain immunoglobulin variable domain that recognises them, conferring antigen specificity and at the same time activating lymphocytes. CARs are structured with a hinge and transmembrane region to anchor the CAR to the T cell membrane and contain intracellular domains to transfer activation signals to the T cell (Figure 1). First generation CARs contained CD3ζ or CD3γ signalling domains alone, but they were not clinically effective [16]. The 2nd and 3rd generation CARs contain at least one costimulatory signalling domain along with CD3ζ. T cells expressing these constructs have demonstrated higher efficacy in treated patients. Some of these additional domains frequently used in these constructs are the costimulatory signalling domains of CD28 and 4-1BB [17]. In fact, in 2003, anti-CD19 2nd generation CAR-T cells provided the first preclinical evidence for antitumor activity against chronic lymphocytic leukemia (CLL) cells, co-stimulated by CD80 and IL-15 [18,19]. Later, 3rd generation CAR-T cells demonstrated better capacities to boost anti-tumour activities by introducing both CD28 and 4-1BB signalling domains [20,21]. Interestingly, a 4th generation of novel CAR-T cells is currently under development, such as armoured CARs and TRUCKs and CAR-T cells with a transgenic payload [22]. Some of these CARs consist of two or three costimulatory or pro-inflammatory signalling domains plus an inducible transgenic cytokine secretion module to activate and confer anti-tumour activity to T cells while immunomodulating the tumour microenvironment [23,24,25]. Other CAR-T cell strategies are exploring their combination with other therapies, such as the combination of two or more CARs with diverse antigen specificities (pooled CAR-T cells) [26], the combination of two or more CARs expressing different tumour antigen specificities within the same lymphocyte (multi-CAR-T cells) [27,28,29], or a single CAR targeting several antigens simultaneously through different antigen-binding domains (tandem CAR-T cells or TanCARs) [30,31,32,33].

CAR-T cell therapies are demonstrating elevated success rates for the treatment of haematological neoplasms [34]. Thus, the only CAR-T cell therapies approved for clinical practice by regulatory agencies are anti-CD19 CAR-T cells for CD19+ B-cell lymphoproliferative malignancies (tisagenlecleucel (Kymriah™), Axicabtagene ciloleucel (Yescarta™), Brexucabtagene autoleucel (Tecartus™), Lisocabtagene maraleucel (Breyanzi™), and anti-BCMA CAR-T cells for multiple myelomas (Idecabtagene vicleucel (Abecma^®^)) [35]. Up to date, 1120 clinical trials are evaluating CAR-T cell biologicals as treatment interventions (483 phase 1, 261 phase 1/2, 93 phase 2, 7 phase 2/3, 12 phase 3, 2 phase 4, and 262 not applicable) (Clinicaltrial.gov, accessed on 18 October 2022). Among them, many clinical trials for solid tumours are being developed with low efficacy and modest results.

To summarise, CAR-T cells effectively enhance and improve T cell effector functions, proliferation activity, anti-tumour efficacy, cytokine secretion and persistence, among other T cell functions [17,36]. Nevertheless, many challenges limit the clinical application of CAR-T cell therapies, especially for solid tumours, due to their acute and severe associated toxicities, wider tumour-antigen heterogeneity, antigen escape, insufficient CAR-T cell persistence and expansion, lack of classical T-attractive chemokines and limited tumour infiltration [37,38]. The use of αβ autologous T cells from a patient’s own pool poses many limitations and hinders the standardisation of protocols such as distinguishing normal cells from malignant cells. The starting cellular products are limited due to the lymphopenia of the patients [39]. In addition, optimal T cell protocols for adequate activation of these cells are associated with the risk of cytokine release storm (CRS) as a result of an overactivation of infused T cells [40]. CAR-T cell therapies have demonstrated to be efficient for the treatment of hematopoietic diseases, but not for solid tumours. This is thought to occur by T cell inhibition within the tumour microenvironment (TME), where the infiltrating T cells face a suppressive environment characterised by low nutrient concentrations, hypoxia and inhibitory cytokines. Moreover, tumour cells often down-regulate MHC expression, which is necessary for the recognition of short peptide antigens by TCR-based engineering [41]. Furthermore, CAR-T has a limited ability to penetrate within the tumour tissue [42]. CAR-T cell trafficking is ineffective because of the immunosuppressive microenvironment and the dense fibrotic matrix present in solid tumours. Finally, antigen escape and solid tumour antigens are two of the most limiting factors. Indeed, recent studies have described the resistance mechanism after CAR-T cell therapy caused by the selection of cancer cells with down-regulated target antigens [43,44]. An additional danger is that a large proportion of tumour-associated antigens are also expressed in normal tissues at varying levels. This can pose a significant threat to CAR-T cell therapies.

To overcome these barriers, novel cutting-edge CAR strategies are being developed, such as the genetic engineering of other key immune cell types such as macrophages, monocytes and other myeloid populations to kill tumours [37]. In the TME, myeloid cells (M) are the innate immune cells with the highest tumour infiltration rate, interacting with other immune cells, mediating immunosuppression, and promoting angiogenesis. Thus, the development of CAR-M adoptive cell therapies could be a good way of delivering cells with anti-tumour potential within the TME.

## 2. Myeloid Cells Sustaining the Tumour Microenvironment

Many studies have highlighted the critical contribution of myeloid cells to the establishment of an immunosuppressive microenvironment that promotes tumour development and hinders therapy efficacy. Myeloid cells constitute the major component of the TME and play a dominant role in suppressing anti-tumour immunity. The most studied tumour-associated myeloid cells (TAMCs) include tumour-associated macrophages (TAMs), dendritic cells (TADCs), neutrophils (TANs) and myeloid-derived suppressor cells (MDSCs). As key players in tumour development, myeloid cells perform a variety of functions in the TME, varying from immune suppression to immune stimulation. Despite the many studies that characterise myeloid cell populations within the TME, their ontogeny and development are still ambiguous, especially for MDSC. This is caused by the phenotypic plasticity of myeloid cells and their unique properties of differentiation. In fact, myeloid phenotypes switch with diverse stimuli. This property can be interestingly rendered as an ideal editable tool for anticancer therapy. TAMs and TANs can acquire anti-tumour activity by differentiating into M1 macrophages and N1 neutrophils. On the other hand, they can become immunosuppressive and tumour-promoting subsets such as M2 macrophages and N2 neutrophils, respectively. This differentiation polarity depends on the factors and cytokines present within TME and produced by a variety of tumour cell types [45,46].

Macrophages are a major component of the TME, and they are thought to dominate the immune landscape within tumours. Macrophages are known as “big eaters”, specialised cell types in the phagocytosis of dead cells and pathogens. Their phenotype is quite plastic and depends on the surrounding cytokines, with M2 being the major phenotype of TAMs. In contrast, M1-like TAMs are involved in activating proinflammatory immune responses as they have a high capacity for antigen presentation. TAMs play a vital role in the occurrence, development, invasion, angiogenesis and metastasis of many malignant tumours [45,47]. Hence, TAMs are strongly associated with poor prognosis. Moreover, TAMs possess significant anti-inflammatory properties and mainly release immunosuppressive factors and cytokines such as IL-10, TGF-*β*, CCL17, CCL18 and CCL22 [47,48,49].

Antigen-presenting cells (APCs), such as dendritic cells (DC), operate as a connection between the adaptive and innate immune systems. As professional antigen-presenting cells, DCs are specialised in the processing of foreign antigens and their subsequent presentation to T cells. Within the TME, TADCs acquire immunosuppressive properties and are characterised by an immature differentiation state coupled with high antigen uptake capacities and inadequate antigen presentation [50]. TADCs can also act by inhibiting tumour-reactive T cells and inducing regulatory T cells (Treg) [51,52,53].

In contrast to the well-known capability of inflammatory neutrophils to eliminate pathogens, activate the immune system, and induce tissue damage in infections, TANs function within the TME as immunosuppressive cells. Several studies suggest that TANs are involved in tumour progression by promoting the angiogenic switch and stimulating tumour cell motility, migration and invasion. Thus, TANs contribute to the establishment of a strongly immunosuppressive microenvironment [54,55].

A major subject of research in immunosuppressive myeloid cells in cancer is the study of MDSC differentiation and function. MDSCs are considered a heterogeneous and phenotypically immature population of myeloid cells that differentiate through a myelopoiesis imbalance, which is enhanced in pathological conditions. MDSCs are classified into two major groups both in humans and mice, namely monocytic MDSCs (M-MDSCs) and granulocytic/polymorphonuclear MDSCs (G-MDSC /PMN-MDSCs). These MDSC subsets phenotypically resemble monocytes and neutrophils, respectively [56]. MDSCs are major promoters of progression and metastasis. These cell types are especially proficient at exerting diverse T cell inhibitory functions. MDSCs are some of the populations that have predictive and prognostic value for patients undergoing cancer treatments [57,58,59].

All these properties indicate that there is a strong interplay between tumour and myeloid cells which is relevant for the establishment of an immunosuppressive TME and promoting cancer development. As these myeloid cells possess particularly good capacities to infiltrate the tumour, these TAMCs constitute attractive targets for reprogramming.

## 3. Modulation of Myeloid Cells as a Therapeutic Strategy

The following section describes ongoing strategies for modulating myeloid cells in preclinical and clinical settings. Several strategies to target TAMCs have been developed taking into consideration their properties, such as their differentiation plasticity and immunosuppressive functions. The most promising strategies to modulate myeloid cells are listed in Appendix A and are categorised by: (1) reprogramming myeloid cells to acquire proinflammatory properties; (2) enhancing the diversity of myeloid cell composition within the TME; (3) functional blockade of immune-suppressive myeloid cells; (4) myeloid cell vaccines; and (5) immunometabolic strategies.

### 3.1. Reprogramming of Myeloid Cells to Acquire Proinflammatory Properties

One approach to reprogram myeloid cells to stimulate anti-cancer activity is to elevate the expression of pattern recognition receptors expressed by APCs. Toll-like receptor 9 (TLR9) agonists are the most extensively studied in human therapy. In fact, preclinical studies on TLR9 agonists demonstrate that these increase antigen-specific T cells in multiple tumour types, including breast cancer, colorectal and non-small-cell lung carcinoma (NSCLC) [60,61,62,63]. Ongoing clinical studies are examining the use of TLR9 agonists in combination with local radiation and immunotherapy (NCT03410901) for the treatment of refractory lymphomas or conventional chemotherapy (paclitaxel and carboplatin) for the treatment of NSCLC [64]. Among the different inhibitors used to polarise immunosuppressive myeloid cells, a phosphatidylinositol-3-kinase γ (PI3Kγ) inhibitor has been used in preclinical murine studies in cancer models. PI3Kγ inhibitors promoted TAM polarisation to a M1 phenotype [65,66]. Additionally, combining eganelisib (PI3Kγ inhibitor) with both anti-CTLA4 and anti-PD-1 therapy results in cure rates of 30%, whereas dual checkpoint inhibition alone did not result in complete responses. These results led to a phase I trial (MARIO-1), which involved eganelisib as monotherapy and in combination with nivolumab (anti-PD-1) for the treatment of advanced solid tumours (NCT02637531).

### 3.2. Targeting the Myeloid Cell Composition within the TME through Enhanced Differentiation, Proliferation and Recruitment

Tumour-derived factors modulate the recruitment, expansion and differentiation of immunosuppressive myeloid to fuel tumour progression. One of the most straightforward strategies for myeloid cell targeting in cancer treatments is the blockade of myeloid chemoattractants. This strategy has been studied in several preclinical animal models and clinical trials, but with unfavourable results. The signalling axis of chemokine (C-C motif) ligand 2 (CCL2)–C-C chemokine receptor type 2 (CCR2) suppresses tumour metastasis through reduced angiogenesis in preclinical models. Inhibition of the CCL2–CCR2 axis reduced the recruitment of immunosuppressive myeloid cells [67,68]. A human monoclonal anti-CCL2 antibody, Carlumab (CNTO 888), had negative clinical results (NCT00992186). In addition, CSF1R inhibitors have been developed without demonstrating meaningful clinical activity in the management of advanced solid tumours (NCT02265536, NCT01346358 and NCT01444404). Interestingly, diverse studies targeting STAT3 have since been developed, including using small-molecule inhibitors (and oligonucleotides (danvatirsen and STAT3 DECOY) [69,70]. Nonetheless, the majority of trials have performed poorly (NCT02753127, NCT02993731 and NCT01839604).

### 3.3. Functional Blockade of Immune-Suppressive Myeloid Cells

Strategies relying on blocking immune escape mechanisms are in development with successful outcomes. SSIRPα and Siglec-10 suppress inflammatory responses in myeloid cells by binding to their respective ligands and maintaining the myeloid immunosuppressive profile in TME. Preclinical studies have shown that blocking CD47/ SIRPα signalling results not only in augmented phagocytosis but also in a decrease in myeloid-driven immunosuppression through M1 macrophage polarisation. Similar results were obtained with anti-CD47/Siglec-10 therapies, resulting in improved phagocytic abilities in macrophages. Antagonist-CD47 has been directly applied to human therapy, and the most promising clinical data is focused on the treatment of relapsed/refractory B cell non-Hodgkin lymphoma (NHL) (NCT02953509). However, antagonist-CD47 has been tested in solid tumours but without satisfactory results (NCT02216409, NCT30811285 and NCT03013218).

### 3.4. Myeloid Cells Vaccines

Cancer vaccines were first developed with limited success until genetic engineering was used to increase the potency of these therapeutic vaccines (Figure 2). Autologous cancer vaccines have been applied to activate DC by combining the use of GM-CSF cell-based vaccines (GVAX) or TEGVAX enhancing antitumoural effects). Moreover, the discovery of tumour-associated antigens (TAAs) allowed the development of specific targeting, including, among others, TAAs overexpressed by cancer cells (GD2, MUC18). certain tissue types (CEA, PSA, Tyrosinase, Gp100, Muc1, and GM2), and antigens unique to germ cells (MAGE, NY-ESO-1, SSX, BAGE, and GAGE) (PMID: 21048000). Interestingly, combinatorial immunotherapeutic regimens in patients with solid tumours using DC immunisation with personalised TAA panels showed TAA immunisation-induced-specific CD4^+^ and CD8^+^ T cells with favourable overall survival (NCT02709616, NCT02808364 and NCT02808416).

### 3.5. Immunometabolic Strategies

Various experimental results indicate that metabolism is an important regulator of immune cell phenotype and function [71,72]. Indeed, metabolic pathways such as glycolysis, oxidative phosphorylation and β-oxidation of fatty acids were up-regulated in myeloid suppressor cells. In fact, the adenosine monophosphate-activated protein kinase (AMPK) pathway is one of the main energy sensors involved in immunosuppressive myeloid metabolism. Metformin activates AMPK, and in vitro studies showed that metformin increases the expression of M1-related cytokines and attenuates M2-related cytokine expression. In addition, metformin decreased the MDSC and M2 macrophage fractions by downregulating the mevalonate pathway [73,74]. Furthermore, several studies have described that metformin modulates ICI immunotherapies in cancer [75,76,77,78].

New strategies need to be developed by understanding the properties of myeloid cells in the tumour.

## 4. New Era of CAR-Engineered Cell Therapies for Innate Immune Cells

In the light of these shortcomings, the substitution of T cells with innate immune cells has emerged as new CAR therapies. These cell types include natural killer (NK) cells, macrophages, dendritic cells, mucosa-associated invariant T (MAIT) cells and innate-like T cells such as γδ T cells and NK T cells. Innate immune cells overcome some of the barriers faced by adoptive immunotherapies, such as antigen-specificity (ADCC) and safety concerns. Indeed, innate cells can kill target cells in a CAR-restricted and unrestricted manner through natural cytotoxicity receptors and antibody-dependent cellular cytotoxicity (ADCC) mechanisms [79]. Moreover, the risk of cytokine release storms (CRS) is lessened [80]. Another advantage is the possibility to use them in allogeneic strategies, whereas CAR-T cells have so far succeeded only as autologous products and face major challenges as allogeneic products. Moreover, innate immune cells such as macrophages and NK have the ability to traffic more efficiently to tumour sites. Finally, CAR expression by innate immune cells reprograms these cells to acquire anti-tumour profiles.

Therefore, the trending topic of innate immune cells for CAR engineering is NK cells and macrophages, which could be an alternative to T cells. In recent years, CAR engineering of natural killer (NK) cells has become attractive due to the NK’s biological features. NK cells have an innate ability to lyse tumour cells, mediated by perforin and granzyme, without any priming [81,82,83]. NK cells also present immunomodulatory functions by releasing cytokines such as IFN-gamma [84]. In fact, NK genetically engineered to express a CAR molecule maintain their intrinsic capacity to recognise tumour cells through their native receptors. In addition, NK cells do not rely on the T cell receptor (TCR) for cytotoxic killing. Moreover, it has been suggested that CAR-NK cells can eliminate immunosuppressive myeloid cells in the TME [85,86]. Additional properties of CAR-NK cells could make them better and potentially safer than T cells. Allogeneic CAR-NK cells should not cause graft-versus-host disease (GVHD), and they can be safely administered without the need for full HLA matching [87]. The risk of toxicity to normal tissues is relatively low due to the limited lifespan of CAR-NK cells in the circulation. In addition, cytokine release syndrome (CRS) and neurotoxicity have not been associated with CAR-NK immunotherapy [88,89]. Furthermore, the expansion of CAR-NK is easier due to several sources for obtaining NK cells. NK can be obtained from immortalised human cell lines, donor peripheral blood, archived samples of umbilical cord blood or differentiation of pluripotent cells [57,90,91]. However, the origin of NK sources limits their maturation stage and viability, which is reflected in the diverse anti-tumour effectiveness of the produced CAR-NK cells. In addition, CAR-NK preparations are shorter in production time than CAR-T [92]. Moreover, NK recognises and kills tumour cells through native receptors independent of the CAR engineering, making it less likely that the disease will escape through down-regulation of the CAR antigens, as shown with CAR-T cell therapy [92,93].

### New Era of CAR-Engineered Cell Therapies for Innate Immune Cells

To date, NK cells have been the most commonly utilised innate immune cell for CAR therapy. The first preclinical applications of CAR-NK cells appeared 17 years ago. Imai C. et al. conducted a study using primary leukemia cells from patients with B-lineage ALL. They developed a novel method that allows specific expansion of NK cells lacking T cell receptors (CD56^+^CD3^−^ cells) and their highly efficient transduction with chimeric receptors (PMC1895123). CD19 is the major target for haematological malignancies [94,95,96]. In contrast, a wide range of tumour antigens have been targeted in solid cancers. Antigens such as EGFR, ErbB2/HER, GD2, Glypican-3 and c-MET have been targeted by CAR-NK for treatment of glioblastoma, breast, ovarian, neuroblastoma, melanoma and liver cancer [91,94,97,98,99]. The main construct used to design CAR-NK is based on the CD19 antigen. The core component of CAR-NK consists of an initial signalling domain (mostly CD3) and a costimulatory domain to form an intracellular signalling motif (mainly CD28, CD137 or 4-1BB). In addition, NK cells used their costimulatory molecules (namely NKG2D and CD244 or 2B4) to increase their cytotoxic capability and cytokine production [100]. Moreover, to improve the cytotoxicity and intratumoural infiltration of CAR-NK, many therapeutics have been tested preclinically in combination with CAR-NK [101,102].

Nowadays, an increasing number of clinical trials have examined CAR-innate immune cells (Figure 3, Appendix A). The first clinical trial with CAR-innate immune preparations in solid tumours has been reported by Xu and colleagues (NCT03294954). Since then, several strategies have been exploited to improve CAR NK therapy traffic to tumours (Heczey and colleagues), demonstrating the ability of NKT cells to traffic to tumours [103]. Other clinical trials have utilised CAR-NK cells directed against tumour antigens such as MUC1 or immune checkpoints such as PD-1 (NCT02839954). In agreement with this, promising preclinical data prompted the launch of three phase I/II clinical trials aimed at assessing the safety and efficacy of ROBO-1-directed CAR-NK cell therapy in PDAC and other solid tumours depending on ROBO-1 expression on cancer cells (NCT03941457, NCT03940820 and NCT03931720). Interestingly, new clinical trials are combining CAR-NK therapy with PD-1/PD-L1 immunoblockade (NCT04847466). Moreover, other combination therapies with expanded and activated unmodified NK cells have already been tested in neuroblastoma and hematopoietic neoplasm studies (NCT02573896, NCT02481934, and NCT02280525).

## 5. Preclinical and Clinical Development of CAR-Innate Immune Cell Strategies

### 5.1. Preclinical Development of CAR-Innate Immune Cell Strategies

Currently, the manipulation of macrophages within the TME has raised great expectations for new treatments. The two main phagocytic cells are monocytes and monocyte-derived macrophages. Considering their antigen presentation capacities and penetration ability in the tumour microenvironment, macrophages should be prioritised for the treatment of solid tumours [104,105]. Engineering macrophages to endow them with anti-cancer properties could be a promising strategy to improve immunotherapies for solid tumours. CAR-macrophage (M) therapies are based on CAR expression in macrophages to enable them to specifically bind tumour-associated antigens and activate their phagocytic activities [106]. In contrast to T cells, macrophages infiltrate more efficiently into solid tumours [99,107]. Currently, one of the key issues in CAR-M research is improving the CAR design for macrophages. Hence, the identification of proteins regulating phagocytosis, macrophage infiltration and antigen presentation is key to selecting appropriate signalling domains to be included in the CARs. Moreover, CAR-M has apparent therapeutic efficacies under some conditions. However, the mechanisms by which CAR-M work and how they will function in the TME are not yet clear. Hence, to apply this approach to human therapy, we need to gain more insight on the mechanisms eliciting anti-tumour immunity by these CAR-expressing macrophages.

CAR constructs for myeloid cell genetic engineering possess the same structure as conventional ones, consisting of extracellular binding-antigen, hinge, transmembrane and intracellular signalling regions. The main difference resides in the intracellular signalling domains used to trigger phagocytic capacities (Table 1). Macrophages express the Syk kinase, which contains tandem SH2 domains that bind to CD3ζ inducing phagocytosis activity [108]. However, like 1st generation CAR-T cells, CD3ζ alone is not sufficient. Thus, other ITAM-containing signalling domains that exert similar functions as CD3ζ to enhance canonical phagocytic and trogocytic signals in macrophages are also being used, such as multiple epidermal growth factor-like domains protein 10 (Megf10), Mer receptor tyrosine kinase (MerTK), and the γ subunit of Fc receptor (FcRγ) [109,110]. CD28, 4-1BB, PI3K, CD86, CD147, TLR2, TLR4 and TLR6 are also being used as intracellular signalling domains [106,109,110,111,112]. CD147 does not enhance phagocytosis but induces T cell infiltration, decreases collagen content, and reduces tumour growth [111]. On the other hand, the addition of anti-CD47 antibodies enhances phagocytosis [106]. Interestingly, Niu et al. enabled macrophages to recognise CCR7+ cells and exert cytotoxic activity through the MerTK cytosolic domain, one of the most efficient phagocytosis inducers in vitro, suppressing tumour growth in vivo with low toxicity, prolonging survival, preventing metastasis, inducing T cell infiltration, and inducing systemic anti-tumour immunity [109].

CT-0508, the first CAR-M to be clinically developed (NCT04660929 clinical trial), reported on its preclinical development through transduction of the human leukemia monocytic cell line THP-1 with an anti-CD19 CD3z-based first-generation CAR [110,113]. Those CAR-Ms demonstrated the phagocytosis capacities of mesothelin or HER2 antigen-positive cells. For the transduction of primary human macrophages, anti-HER2 CAR-Ms were generated via Ad5f35 transduction, a replication-incompetent chimeric adenoviral vector. These procedures polarised M0 (uncommitted) macrophages into M1 (pro-inflammatory) macrophages with efficient CAR expression and macrophage activation. This CAR-M demonstrated good therapeutic capacities by reducing tumour growth and improving survival through the modulation of the TME and the induction of systemic T cell responses (including CD4, CD8, and IFNγ TILs) [110,113]. Moreover, CT-0508 caused abscopal effects against HER2 tumours and protected mice from tumour recurrence.

### 5.2. Clinical Development of CAR-Myeloid Strategies

Three clinical trials are studying CAR-myeloid strategies (Appendix A) (Figure 4).

#### 5.2.1. NCT04660929

This is an interventional first-in-human phase I open label study of adenovirally transduced autologous anti-HER2 CAR macrophages in human growth factor receptor 2 (HER2) overexpressing solid tumours. The studied therapy is CT-0508, a CAR-engineered macrophage developed by Carisma Therapeutics. This trial studies a U-S- FDA-regulated drug product. The estimated study completion date is February 2023. Two groups of patients are included, both receiving the full dose manufactured per patient. Group 1 will undergo intra-subject dose escalation of IV administrations of up to 500 million total cells on day 1, up to 1.5 billion total cells on day 3, and up to 3.0 billion total cells on day 5. Group 2 will receive the full dose IV on day 1 of up to 5 billion cells. The primary outcome measures include the safety and tolerability of CT-0508 by estimating the frequency and severity of adverse events, the frequency and severity of Cytokine Release Syndrome (CRS) (for 14 months) and assessing the feasibility of manufacturing CT-0508. The secondary outcome measures include the objective response rate (ORR) according to RECIST v1.1 of at least 1 dose of CT-0508 among subjects with HER2 overexpressing solid tumours; the proportion of subjects with an OR (either a complete response [CR] or partial response [PR]) in subjects who received at least 1 dose of CT-0508 and at least the 8-week tumour evaluation as determined by the investigator using RECIST v1.1 (for 24 months); and progression-free survival (PFS) (for 24 months) defined as the time between the date of the first dose and the date of the first documented disease progression by the investigator using RECIST v1.1 or death, whichever occurs first. The inclusion criteria include HER2-positive recurrent or metastatic solid tumours for which there are no available curative treatment options. Breast cancer and gastric/gastroesophageal junction cancers must have failed to respond to approved HER2-targeted agents. Other HER2-positive tumour types must have failed standard of care therapies, while prior therapy with anti-HER2 drugs is not required. These tumours include adenocarcinoma, bile duct cancer, biliary tract cancer, bladder cancer, breast cancer, lung cancer, ovarian cancer, colorectal cancer, stomach neoplasms, pancreatic cancer and endometrial cancer, among others. Other inclusion criteria include Eastern Cooperative Oncology Group (ECOG) performance status 0 or 1 and adequate bone marrow and organ function. Exclusion criteria include HIV, active hepatitis B or hepatitis C infection, a diagnosis of immunodeficiency or chronic exposure to systemic corticosteroid therapy or any other form of immunosuppressive therapy, untreated or symptomatic central nervous system (CNS) metastases or cytology-proven carcinomatous meningitis and a left ventricular ejection fraction (LVEF) <50% as determined by ECHO or a multiple gated acquisition scan (MUGA).

Remarkably, the pharmaceutical company is planning to initiate a clinical trial of CT-0508 in combination with anti-PD-1 (Pembrolizumab).

#### 5.2.2. NCT05007379, the CARMA-2101 Study

This is an observational clinical trial developed by the Centre Oscar Lambret with a cohort study to assess the anti-tumour activity of new CAR-M in breast cancer patients’ derived organoids (CARMA). This is a cohort study conducted in breast cancer patients who require a surgery or a tumour biopsy as part of their care and does not study a U.S. FDA-regulated drug product. Other biological samples will also be collected, such as blood to analyse the host’s inflammatory status. For primary outcome measures, the anti-tumour activity of the CAR-macrophages is compared against organoids from HER2 negative, HER2 low and HER2 positive breast cancers (for 24 months) and to non-modified macrophages (for 24 months). As a secondary outcome measure, the anti-tumour activity of the CAR-M is compared against organoids from early and advanced breast cancer patients (for 24 months). The inclusion criteria include histologically confirmed breast cancer at any stage, requiring surgery or a tumour biopsy as standard of care, any or no systemic treatment and >18-year-old adults. 100 enrolled participants are estimated for this prospective study. The estimated study completion date is 1 September 2023.

#### 5.2.3. NCT05138458, the IMAGINE Study

IMAGINE is a phase I/II multicentre, open-label, dose-escalation and dose cohort expansion clinical trial evaluating MT-101 treatment in patients with CD5+ refractory or relapsed T cell lymphoma. It is a multi-ascending dose escalation, sequential assignment, non-randomised trial. Its estimated study completion date is October 2024. MT-101 is engineered with myeloid cells derived from the patient’s blood and administered in two arms. The first one receives the MT-101 biological (CD5 AKAT cells) (cohort 1 and 3), and the second one receives MT-101 preceded by conditioning lymphodepleting chemotherapy (IV administration of fludarabine and cyclosphosphamide) [114]. In the first part of the study, cohort 1 and 2 will receive a low dose of cells, and cohorts 3 and 4 will receive a higher dose of cells, to assess safety and tolerability. In the second part of the study, cells with or without chemotherapy will be administered based on results of Part 1 and the safety, tolerability, and efficacy of MT-101 will be assessed. All patient groups will receive six doses of the drug product over 3 weeks. The primary outcome measures include safety and tolerability based on observed adverse events (AEs), including all potential dose limiting toxicities (for 4 weeks). Secondary outcomes include MT-101 cell kinetics in blood, measured by the quantity of its RNA in the blood (for 4 weeks); the objective response rate, defined as the percentage of subjects achieving the best overall response of complete response (CR) or partial response (PR) (for 24 weeks). Other outcome measures include duration of response (DOR), defined as the date of first assessment of PR or CR to the date of the follow-on first documentation of progressive disease or death, whichever occurs earlier; progression-free survival (PFS), defined as the time from the date of the first administration of MT-101 to the date of first documentation of progressive disease or death, whichever occurs earlier; and overall survival (OS), defined as the time from the date of the first administration of MT-101 to the date of death (all for 48 weeks). Key inclusion criteria include refractory or relapsed pathologically confirmed T Cell Lymphoma (TCL): peripheral T cell Lymphoma not otherwise specified (PTCL-NOS), angioimmunoblastic T cell Lymphoma (AITL), ALK-negative anaplastic large cell lymphoma (ALCL), ALK-positive ALCL, or mycosis fungoides (MF) stage IIB-IV including large cell transformation; CD5-expressing tumours by IHC or flow cytometry of tumour biopsy within 3 months of screening or at screening, Eastern Cooperative Oncology Group performance status <2, adequate organ function as defined in the protocol and being an adults age > or equal to 18 at the time the informed consent is signed. Key exclusion criteria include B1 and B2 disease (as defined in the protocol for subjects with MF), known central nervous system involvement by PTCL, history of allogeneic transplant, history of intolerance to leukapheresis, plasmapheresis, or blood donation, pregnant or nursing women, any acute illness including fever (>100.4 °F or >38 °C), except fever related to tumour, active systemic bacterial, fungal, or viral infection, active chronic infection, other primary malignancies, except adequately treated malignancies or complete remission, active autoimmune disease that has required systemic therapy in the last 2 years, history of hemophagocytic lymphohistiocytosis, history of severe, immediate hypersensitivity reaction attributed to penicillin, any other condition that, in the opinion of the Investigator, would make the subject unsuitable for the study or unable to comply with the study requirements.

## 6. Conclusions and Key Ideas

Adoptive T cell therapy (APC) has emerged as a promising therapeutic strategy to treat cancer that might overcome the clinical limitations of immune checkpoint blockade immunotherapies.CAR-T cell therapies are demonstrating success for the treatment of haematological malignancies, but these therapies fail for the treatment of solid tumours. In order to overcome these critical barriers, novel cutting-edge CAR strategies are being developed, such as the genetic engineering of other key innate immune cell types such as NK cells, macrophages, monocytes and other myeloid populations.CD3ζ, CD28, 4-1BB, Megf10, MerTK, PI3K, CD86, CD147, TLR2, TLR4 and TLR6 are being used as intracellular signalling domains for CAR-M generation, to enhance its phagocytic capacities.Advanced strategies need to be developed by understanding the properties of myeloid cells in the tumour. A clear example is multi-omics data analysis that provides insights into myeloid cell phenotypes.CAR-M has been demonstrated in vitro and in vivo to suppress tumour growth in vivo with low toxicity, prolong survival, prevent metastasis, induce T cell infiltration, and induce systemic anti-tumour immunity.In the new CAR-M generation, the adaptation of cytokine receptor domains is required to further improve the immune modulation and tumour killing capabilities of CAR-M products.CAR-M therapies are being developed in three clinical trials.CAR-NK therapies are being developed in 41 clinical trials.While recent works have demonstrated the feasibility of using CAR-immune innate cells as a therapeutic product, the mechanisms by which they function in the TME and how this may lead to tumour rejection remain unclear. To utilise this approach, we need to understand more about how CAR-immune innate cells function within tumours and can be manipulated to generate.

## Figures and Tables

**Figure 1 biomedicines-10-03035-f001:**
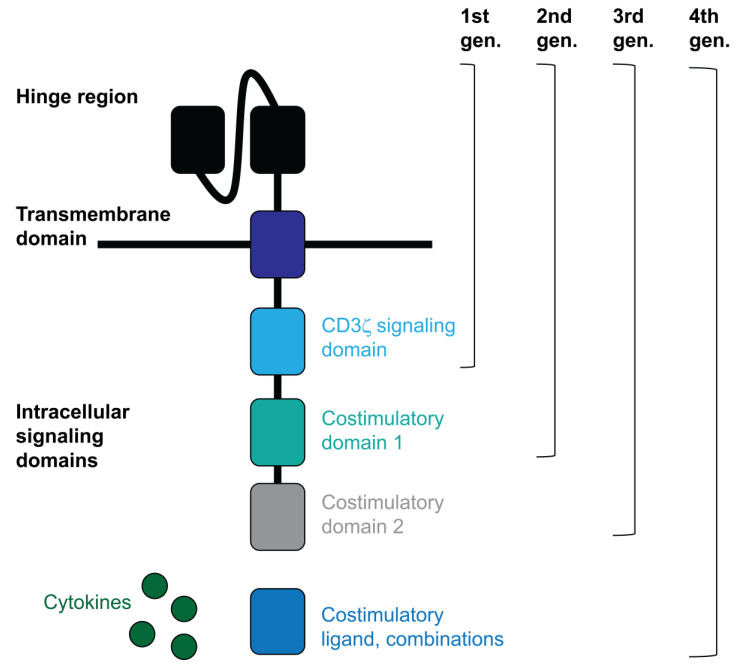
CAR structure.

**Figure 2 biomedicines-10-03035-f002:**
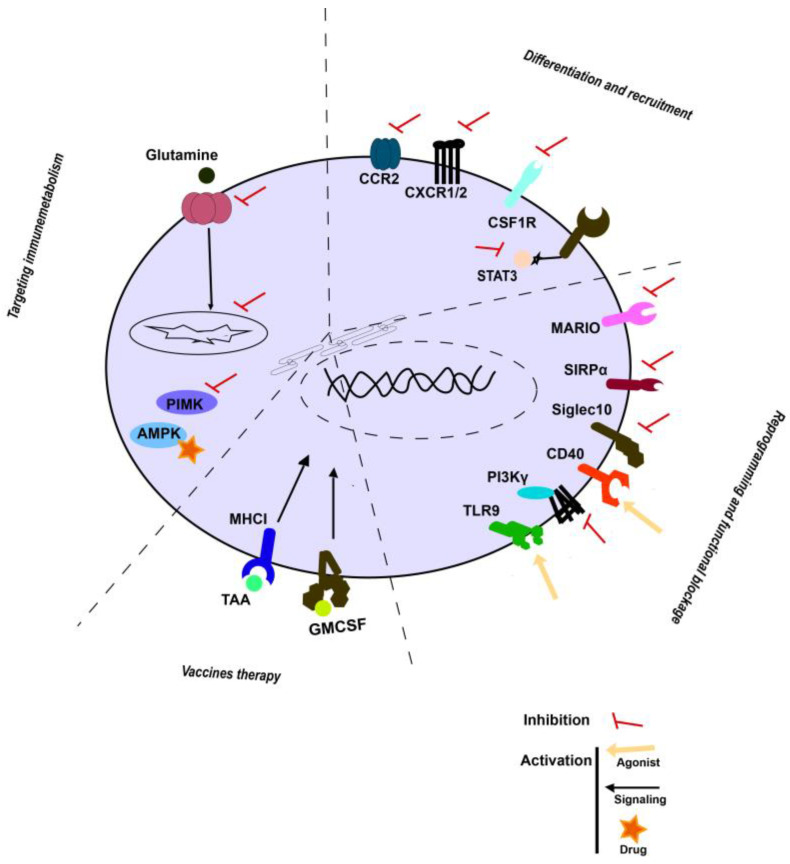
Emerging strategies in targeting TAMCs. Schematic representation of main strategies described above.

**Figure 3 biomedicines-10-03035-f003:**
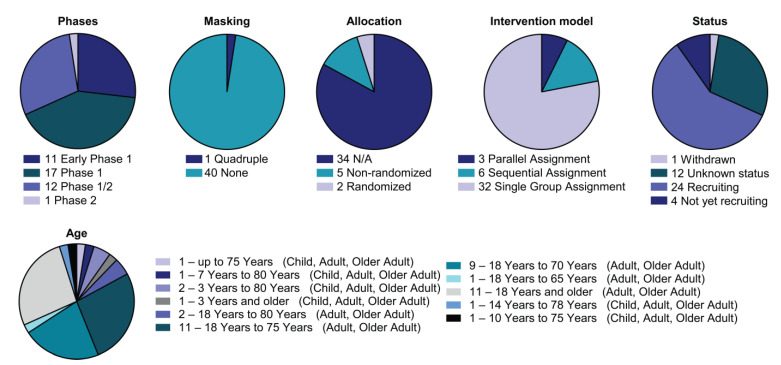
Detailed analysis of the CAR-NK clinical landscape. Pie charts with categorisation of CAR-NK clinical trials by status and study design as indicated (phase, masking, allocation, intervention model assignment, status and age). All current CAR-NK trials are interventional.

**Figure 4 biomedicines-10-03035-f004:**
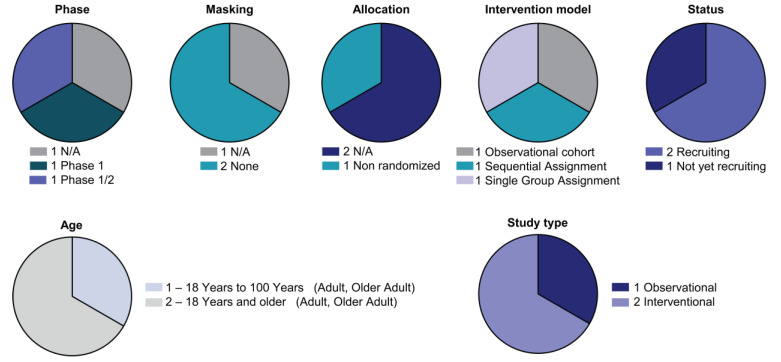
Detailed analysis of the CAR-M clinical landscape. Pie charts with categorisation of CAR-NK clinical trials by status and study design as indicated (phase, masking, allocation, intervention model assignment, status, age and study type).

**Table 1 biomedicines-10-03035-t001:** CAR-T, CAR-NK and CAR-M signalling domains. Main domains applied in CAR constructions.

CAR-T	CAR-NK	CAR-M
CD3ζ	CD3ζ	CD3ζ
CD3γ	CD28	CD28
CD28	4−1ΒΒ	4−1ΒΒ
4−1ΒΒ	CD137	Megf10
OX40	2B4	MerTK
CD20	DAP10	PI3K
CD137	DAP12	CD86
ICOS	NKG2D	CD147
CD40	CD244	TLR2
CD27	Etc.	TLR4
Etc.		TLR6

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
