# Peer review of "Cutting-Edge CAR Engineering: Beyond T Cells"

_biomedicines, 2022, doi:10.3390/biomedicines10123035_

Round 1

Reviewer 1 Report

In this review, Luisa and co-workers give a summary on the basis and clinical results of CAR-myeloid cell immunotherapies. The development of CAR-myeloid cell therapies could be a good way of delivery cells for potentiating immunotherapeutic effect against solid tumors. I think this review would generate broad interest for researchers in the field of onco-immunology. A revised manuscript is recommended for publication in Biomedicines.

1.     It is suggested to provide a schematic illustration on myeloid cell types and mechanisms for section 3.

2.     How about combine section 5 and section 6?

3.     In the last section, the limitations, challenges, and future development direction for clinical translation of CAR-myeloid engineering should be concluded.

4.   All abbreviations should be given full names where they first appear.

Author Response

We sincerely appreciate the comments from Reviewer 1, and we thank the Reviewer for the points to improve. We have addressed all the issues as follows, point by point:

In this review, Luisa and co-workers give a summary on the basis and clinical results of CAR-myeloid cell immunotherapies. The development of CAR-myeloid cell therapies could be a good way of delivery cells for potentiating immunotherapeutic effect against solid tumors. I think this review would generate broad interest for researchers in the field of onco-immunology. A revised manuscript is recommended for publication in Biomedicines.

  1. It is suggested to provide a schematic illustration on myeloid cell types and mechanisms for section 3.

A schematic illustration on myeloid cell types and mechanisms has been included (Figure 2).

  1. How about combine section 5 and section 6?

Sections 5 and 6 have been combined.

  1. In the last section, the limitations, challenges, and future development direction for clinical translation of CAR-myeloid engineering should be concluded.

The section tittle has been changed to "Clonclusions and key points"

  1.  All abbreviations should be given full names where they first appear.

Abbreviations have been carefully revised and given full names where they first appear.

Reviewer 2 Report

Luisa Chocarro and colleagues submitted a well-written review manuscript describing cutting-edge CAR engineering beyond T cells.

Authors summarize the basic, translational, and clinical results of CAR-innate immune cells and CAR-M cell immunotherapies from their engineering and mechanistic studies to preclinical and clinical development. 

Authors cover such aspects as adoptive T cell therapy, myeloid cells sustaining the tumour microenvironment, modulation of myeloid cells as a therapeutic strategy, new era of CAR-engineered cell therapies for innate immune cells, preclinical development of CAR-innate immune cell strategies, clinical development of CAR-myeloid strategies.

Finally, authors sugges that while recent works have demonstrated the feasibility of using CAR-immune innate cells as a therapeutic product, the mechanisms by which they function in the TME and how this may lead to tumour rejection remain unclear. To utilize this approach, scientists need to understand more about how CAR-immune innate cells function within tumours and can be manipulated to generate.

Overall, authors present a quality and well-written manuscript valuable for the scientific community and should be accepted for publication after minor edits are made.

============

Other comments:

1) Please check for typos and punctuation throughout the manuscript.

2) Authors are kindly encouraged to cite this article that describes various aspects of CAR-T functioning, including knowns and unknowns factors that affect CAR-T cell dysfunction. DOI: 10.3390/cancers14041078

Author Response

We sincerely appreciate the comments from Reviewer 2, and we thank the Reviewer for the points to improve. We have addressed all the issues as follows, point by point:

Luisa Chocarro and colleagues submitted a well-written review manuscript describing cutting-edge CAR engineering beyond T cells.

Authors summarize the basic, translational, and clinical results of CAR-innate immune cells and CAR-M cell immunotherapies from their engineering and mechanistic studies to preclinical and clinical development. 

Authors cover such aspects as adoptive T cell therapy, myeloid cells sustaining the tumour microenvironment, modulation of myeloid cells as a therapeutic strategy, new era of CAR-engineered cell therapies for innate immune cells, preclinical development of CAR-innate immune cell strategies, clinical development of CAR-myeloid strategies.

Finally, authors sugges that while recent works have demonstrated the feasibility of using CAR-immune innate cells as a therapeutic product, the mechanisms by which they function in the TME and how this may lead to tumour rejection remain unclear. To utilize this approach, scientists need to understand more about how CAR-immune innate cells function within tumours and can be manipulated to generate.

Overall, authors present a quality and well-written manuscript valuable for the scientific community and should be accepted for publication after minor edits are made.

============

Other comments:

1) Please check for typos and punctuation throughout the manuscript.

The manuscript has been carefully revised.

2) Authors are kindly encouraged to cite this article that describes various aspects of CAR-T functioning, including knowns and unknowns factors that affect CAR-T cell dysfunction. DOI: 10.3390/cancers14041078

The reference has been added (ref. 10).

Reviewer 3 Report

well written review

Author Response

We sincerely appreciate the comments from Reviewer 3.